# Students distracted by electronic devices perform at the same level as those who are focused on the lecture

Romesh P. Nalliah[1] and Veerasathpurush Allareddy[2]

[1] Department of Global Health, Harvard School of Dental Medicine, Boston, MA, USA
[2] Department of Orthodontics, College of Dentistry, The University of Iowa, Iowa City, IA, USA

## ABSTRACT

**Background.** Little is known about the characteristics of internet distractions that students may engage in during lecture. The objective of this pilot study is to identify some of the internet-based distractions students engage in during in-person lectures. The findings will help identify what activities most commonly cause students to be distracted from the lecture and if these activities impact student learning.

**Methods.** This study is a quasi-experimental pilot study of 26 students from a single institution. In the current study, one class of third-year students were surveyed after a lecture on special needs dentistry. The survey identified self-reported utilization patterns of "smart" devices during the lecture. Additionally, twelve quiz-type questions were given to assess the students' recall of important points in the lecture material that had just been covered.

**Results.** The sample was comprised of 26 students. Of these, 17 were distracted in some form (either checking email, sending email, checking Facebook, or sending texts). The overall mean score on the test was 9.85 (9.53 for distracted students and 10.44 for non-distracted students). There were no significant differences in test scores between distracted and non-distracted students ($p = 0.652$). Gender and types of distractions were not significantly associated with test scores ($p > 0.05$). All students believed that they understood all the important points from the lecture.

**Conclusions.** Every class member felt that they acquired the important learning points during the lecture. Those who were distracted by electronic devices during the lecture performed similarly to those who were not. However, results should be interpreted with caution as this study was a small quasi-experimental design and further research should examine the influence of different types of distraction on different types of learning.

# INTRODUCTION

Most students currently enrolled in dental schools in the United States (US) were born in the 1980s or 1990s (*Zickuhr, 2010*; *Elam, Stratton & Gibson, 2007*). This generation is referred to as Generation Y (Gen Y) and they function very differently to previous

Corresponding author
Veerasathpurush Allareddy,
Veerasathpurush-
Allareddy@uiowa.edu

generations of dental students. Research in dental hygiene education has shown that Gen Y students revel in group work and are sagacious technology users (*Blue, 2009*).

Previous research has shown that passively listening to lectures is less effective than being engaged in a lecture where the student must solve "retrieval" questions that require them to go back to the information and find the answers (*Karpicke & Blunt, 2011*). The flipped classroom model is based on this concept, and retrieval questions and discussions are during the classroom session. Gen Y students also have a proclivity to multitask and a need for immediate feedback, which the retrieval questions would provide (*Blue, 2009*). It is not currently known if multitasking during lectures impacts learning outcomes. Lectures are designed to be uni-tasking experiences that require the student to be fully engaged in the verbal and (sometimes) visual dissemination of information. Traditional lectures do not support multitasking activities and may actually be in conflict with them.

A recent study on the impact of lecture retention from fidgeting and mind-wandering showed that retention of lecture material declined as time spent on a task increased. Additionally, fidgeting also increased as time increased, and fidgeting had a negative impact on retention (*Farley, Risko & Kingstone, 2013*). This suggests that shorter time-span educational activities may be more effective for retention. This paper reports the outcomes of a small quasi-experimental pilot study that assessed the role of distraction versus not in learning outcomes of a single lecture at Harvard School of Dental Medicine (HSDM). The post-lecture test was designed to measure content retention from the lecture and the post-lecture survey is designed to determine what electronic activities, like texting and emailing, students were engaged in. The objective of the paper is to compare retention outcomes to level of engagement in electronic activities. Although engaging in electronic activities doesn't necessarily mean the student was distracted, we have compared whether or not students engaged in these activities to their performance on a content-retention test.

## MATERIALS AND METHODS

### Study design
The current study is a pilot cross-sectional study at Harvard School of Dental Medicine. A traditional lecture (on special needs dentistry) was delivered and a post-lecture questionnaire was administered to a 3rd-year class at HSDM. After the traditional lecture, a post-lecture test including 12 multiple choice test questions relating to the lecture content was administered to evaluate how effectively students learned the information in the lecture. The post-lecture test measured understanding and knowledge of the important concepts from the lecture. The post-lecture survey (this was not anonymous) also asked several questions about what electronic activities students were engaged in during the lecture. Harvard Medical School Institutional Review Board exemption was acquired for this study. The protocol number is IRB13-1300.

### Analytical approach
Simple descriptive statistics were used to summarize the data. The primary outcome variable was the test scores (computed on a scale of 0 to 12). There were 12 multiple

choice questions and each question was assigned one point. The primary independent variables of interest were if the students were distracted, type of distraction, and gender. The distribution of test scores was compared between distracted and non-distracted students using Mann-Whitney U tests. Two multivariable linear regression models were used to examine the simultaneous influence of gender and distraction on test scores. In the first model, a composite variable (whether the student was distracted by any form) for distraction and gender were used as independent variables. In the second model, the different types of distractions (checking email, sending email, checking Facebook, or sending text) were used separately along with gender. In both models, the test scores were the outcome variable. Both regression models were fit using the Ordinary Least Squares approach. All tests were two sided and a $p$-value of $<0.05$ was deemed to be statistically significant. All statistical analyses were conducted using SPSS Version 22.0 software (IBM Corp, Research Triangle Park, NC).

## RESULTS

There were 27 students who participated in this lecture. One student did not complete the test or survey and was omitted from the evaluation leaving a final sample of 26 students. The final sample was comprised of 8 males and 18 females.

During the lecture, 57.7% reported that they checked their email and 11.5% reported sending an email, 15.4% checked Facebook, and 7.7% sent a text message. Of those who checked their email, 69% used their smartphone, 18% used their laptop and 13% used an iPad. Seventeen of the 26 students (65.4%) were distracted in some form during the lecture.

There were 12 post-lecture multiple choice questions related to the lecture materials and the proportion of the class that got each question correct is listed: Q1, 73%; Q2, 65%; Q3, 69%; Q4, 96%; Q5, 81%; Q6, 81%; Q7, 85%; Q8, 92%; Q9, 96%; Q10, 58%; Q11, 88%; Q12, 100%. On a scale of 0 to 12 (with 1 point for each question), the mean score for the class was 9.85 (standard deviation is 2.89). The distribution of test scores are summarized in Table 1. The mean score amongst those that were distracted was 9.53 (compared to 10.44 for students that were not distracted. Overall, there was no significant difference in distribution of test scores between students that were distracted and not distracted ($p$-value from Mann-Whitney U test is 0.652). In the survey, 100% of students believed that they understood all the important points from the lecture.

Results of the multivariable linear regression analysis examining the simultaneous association of gender and distraction on test scores are summarized in Table 2. After adjustment for the effects of distraction, males were associated with 2.6 points higher scores compared to females. However, this was not statistically significant ($p = 0.052$). After adjustment for the effects of gender, those who were distracted did not have a significantly different score when compared to those who were not distracted ($p = 0.955$). Gender and distraction explained 17.5% of variance in test scores.

Results of the multivariable linear regression examining the effects of different types of distractions and gender on test scores are summarized in Table 3. Overall, there were
**Table 1  Distribution of Scores (Scale of 0 to 12).**

| Measure | All students | Students that were distracted ($N = 17$) | Students that were not distracted ($N = 9$) | *p*-value |
|---|---|---|---|---|
| Mean | 9.85 | 9.53 | 10.44 | 0.652 |
| Standard deviation | 2.89 | 3.20 | 2.24 | |
| Minimum | 4 | 4 | 6 | |
| 25th percentile | 7.75 | 6 | 8.50 | |
| Median | 12 | 12 | 12 | |
| 75th percentile | 12 | 12 | 12 | |
| Maximum | 12 | 12 | 12 | |

Notes.
 Mann-Whitney U test was used to compare distribution of scores between distracted and non-distracted students.

**Table 2  Multivariable linear regression analysis for examining the effects of gender and distraction on test scores.**

| Independent variable | Parameter estimate (95% CI) | *p*-value |
|---|---|---|
| **Gender** | | |
| Male | 2.60 (−0.02–5.21) | 0.052 |
| Female | Reference | |
| **Distracted** | | |
| Yes | 0.07 (−2.47–2.61) | 0.955 |
| No | Reference | |

no statistically significant differences in test scores for the different types of distractions: checking email (estimate is −0.88, $p = 0.476$), sending email (estimate is 2.40, $p = 0.166$), checking Facebook (estimate is −2.16, $p = 0.293$), or sending text (estimate is 3.66, $p = 0.199$) after adjusting for the effects of gender (estimate is 2.44, $p = 0.007$). In this model, gender and different types of distractions explained 32% of variance in test scores.

## DISCUSSION

Lecture theaters used to be the only source of information but education is moving in the direction of readily available information that is convenient and accessible at all times through electronic resources. Current generations of students are thought to require more engaged teaching modalities (*Massachusetts Institute of Technology, 2014*). In fact, a Pew report found that 87% of teachers believed modern technology was creating an easily distracted generation of students with short attention spans (*Purcell et al., 2012*). Another Pew study showed that 24% of Gen Y report that technology use is what makes their generation unique (*Pew Research Center, 2010*). However, little is known about the impact on learning of being engaged in electronic activities during lecture. This paper reports outcomes of a small study that was designed to evaluate the learning outcomes of a traditional lecture among Gen Y students.

In the current study, students attended a traditional lecture and were given a post-test about the lecture topic and a questionnaire. The questionnaire asked the students whether

Table 3 Multivariable linear regression analysis for examining the effects of gender and different types of distractions on test scores.

| Independent variable | Parameter estimate (95% CI) | p-value |
|---|---|---|
| **Gender** | | |
| Male | 2.44 (−0.23–5.10) | 0.07 |
| Female | Reference | |
| **Distracted by checking email** | | |
| Yes | −0.88 (−3.39–1.64) | 0.476 |
| No | Reference | |
| **Distracted by sending email** | | |
| Yes | 2.40 (−1.08–5.87) | 0.166 |
| No | Reference | |
| **Distracted by checking Facebook** | | |
| Yes | −2.16 (−6.32–2.01) | 0.293 |
| No | Reference | |
| **Distracted by sending text** | | |
| Yes | 3.66 (−2.08–9.39) | 0.199 |
| No | Reference | |

they checked/sent email, checked Facebook accounts or sent text messages during the lecture. Certainly, there is a possibility that students were not completely honest with their answers and our findings may be an underrepresentation of the actual amount of involvement with electronic devices and the Internet that was unrelated to the lecture. We found that 57.7% of students checked their email; 11.5% sent an email; 15.4% checked their Facebook accounts and 7.7% sent a text message during the lecture. Remarkably, the "distracted" group (those that engaged in one of these activities during the lecture) performed similarly in the post-test to the undistracted group.

A total of 65.4% of the class reported engaging in "distracting" behavior such as emailing, using Facebook or texting. Nonetheless, 100% of the students believed that they had understood the important concepts discussed in the lecture. However, in some questions, only 58% of students knew the correct answer. The major concern is that all students believed they understood the important concepts but there were three questions in the post-test where less than 70% knew the correct answer. Overall learning outcomes were not ideal, however, the group that reported being distracted performed similarly to the group that said they were not distracted. Existing research concurs with this finding and reports that media multitasking was not related to self-reported difficulties in distractibility (*Pew Research Center, 2010*). In the current study, 58.8% of the "distracted" group and 55.6% of "non-distracted" answered all questions correctly.

Notably, in the current study all males who were engaged in a "distracting" behavior scored 100% in the post-test. However, among females engaged in "distracting" behaviors only 50% got all questions correct. The mean score for males was 11.63 points compared to 9.06 in females. However, the overall scores were not statistically significantly different between males and females. The small sample size in the current study could have

precluded us from identifying a statistically significant difference in test scores between males and females. It should also be noted that non-distracted males also performed better than their female counterparts and there may be some gender bias in our test. Additionally, our pilot study is small and there is insufficient statistical power to demonstrate that men multitask better during dental school lectures, however, it is interesting that males seemed to outperform females when "distracted" during the current study. This finding is in conflict with several articles in the media but concurs with one previous scientific study (*Palermo, 2014a*; *Palermo, 2014b*; *DeLuca, 2014*; *Mäntylä, 2013*). It may be possible that multitasking during a lecture does not significantly affect learning among males but does reduce learning among females. Larger studies are necessary to evaluate this further.

It is, however, important to note that the overall mean test score for distracted students was 9.53 (on a scale of 0 to 12), whereas, the mean score for non-distracted students was 10.44. Measures that reduce distracted behaviors such as blocking wireless internet access in lecture theaters may aid in maintaining the effectiveness of lectures as a mode of education in the modern era.

An interesting study comparing emergency room (ER) doctors to regular ward doctors found that ER doctors switched tasks more frequently. However, ward doctors multitasked more frequently than ER doctors (*Walter et al., 2014*). It seemed from the study that safety may be implicit in task-switching and multitasking decisions. In the current study of dental students, we found that multitasking didn't necessarily have a negative impact on learning performance as some distracted students (particularly males) were able to score 100% in the post-test.

Since current generations of students are very comfortable with technology and often have their electronic device near them, some thought should be given to the integration of these devices as learning tools for medical and dental students as they transition to independent practice. More research is necessary to evaluate patient perception of electronic device use by doctors and the merits of including appropriate use of electronic devices during education and patient visits.

Additionally, caution should be used when embracing new methods of teaching. The current study shows that students who became distracted during a traditional lecture performed similarly to those who were not. Educational outcomes and costs to the institution should be thoroughly considered when implementing curriculum changes. Larger studies that compare educational outcomes of traditional lectures to other modalities of teaching will help determine the place of the traditional lecture in modern curriculum.

It should be noted that this was a small, quasi-experimental, exploratory study and only provides basic pilot data. The study could be underpowered to find statistically significant associations. A larger study that evaluates students over a semester of lectures and evaluates electronic activity without seeking accurate self-reporting is needed to confirm results. Another limitation is that students answered the post-lecture test and then, immediately, answered the post-lecture survey which requires them to self-report activities related to electronic activities. This may result in biased answers as students realize the purpose of the surveys. The regression models in the current study explained only a small proportion of

variance in test scores. This clearly shows that there could be a multitude of other variables apart from gender and distractions which could influence test scores. Consequently, the issue of omitted variables bias should not be discounted. Finally, this is a small, single site study and information may not be generalizable.

## CONCLUSIONS

Sixty five percent of students in a traditional lecture reported being distracted by email, Facebook or text messages. Those who were distracted during the lecture performed similarly in the post-lecture test to the undistracted group. However, results should be interpreted with caution as this study was a small quasi-experimental design and further research should examine the influence of different types of distraction on different types of learning.

### Funding
This study was not funded.

### Competing Interests
Veerasathpurush Allareddy is an Academic Editor for PeerJ.

### Author Contributions
- Romesh P. Nalliah conceived and designed the experiments, performed the experiments, analyzed the data, contributed reagents/materials/analysis tools, wrote the paper, prepared figures and/or tables, reviewed drafts of the paper, gave final approval of the manuscript.
- Veerasathpurush Allareddy conceived and designed the experiments, analyzed the data, contributed reagents/materials/analysis tools, wrote the paper, prepared figures and/or tables, reviewed drafts of the paper, gave final approval of the manuscript.

### Human Ethics
The following information was supplied relating to ethical approvals (i.e., approving body and any reference numbers):

Harvard School of Dental Medicine - Harvard Medical School exempt status was obtained (IRB #: 13-1300).

### Supplemental Information
Supplemental information for this article can be found online at http://dx.doi.org/10.7717/peerj.572#supplemental-information.

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
