# Peer review of "Students distracted by electronic devices perform at the same level as those who are focused on the lecture"

_PeerJ, doi:10.7717/peerj.572_

## Round 0.1 · original submission · Major Revisions

Dear Authors,

Thank you for submitting a paper on an important topic. Despite my enthusiasm for this paper, there are numerous issues at this time. I concur with both reviewers but believe if significant changes are made we can view a resubmission. Some of the most pressing issues include:

The background does not explore the numerous research on actual distraction in classrooms but rather explores alternative teaching methods which have no relevance to the data. A detailed description of distraction in the classroom with technology is suggested.

The method of data collection is a problem with regard to self-report following the lecture and all limitations of the method highlighted by reviewers should be accounted for in the manuscript – especially in the limitations. Including all other questions about attention if collected would strengthen the paper.

The analyses were limited and did not take advantage of all the methods of analysis that could be completed with this data such as correlations and regressions. A completely new analysis section should be written

The discussion makes some global inferences not supported by the paper and does not address the previous conflicting literature. This should be addressed in detail.

I recommend the authors use terms like quasi-experimental, pilot and exploratory initial often in this paper.

These revisions and the revisions proposed by the other reviewers constitute a major overhaul of this paper. Despite one reviewer rejecting it outright, if the significant changes are made, I will re-review this paper as it is an important topic.

·

Basic reporting

An interesting study and clear written paper! However, I have several suggestions:
#1 Knowing whether a person uses electronic devices during classroom is not the same as knowing the person was distracted. Authors should discuss this aspect and incorporate prior literature.
#2 Authors should instead leave out all remarks/background concerning new, alternative teaching methods as their study focused on a tradiditional lecture. (e.g., it is not necessary to mention the Khan Academy).

Experimental design

#1 In my opinion it is a problem that the authors chose a cross-sectional design. It would be better to ask students after every single lecture in a semester whether they used electronic devices during that lecture and the assess performance in the topic of the lecture in the end of the semester.
#2 Furtmermore, the post-session questionnaire can be problematic as students answer questions on aspects they learned (or should have learned) during the lecture AND questions regarding their engagement of electronic activities during the lecture at the same time. Thus, students might form their personal hypothesis on the research question that was focused on with the questionnaire and answers might be biased.
#3 Were there any control questions (e.g., How distracted did you feel?) -- see #1 in the basic reporting area.

Validity of the findings

#1 The findings/results would be much more valid if engagement in electronic activities could be assessed objectively and/or long-term...
#2 As I mentioned in the experimental design section, it could be that results are biased -- the result that males who were distracted outperformed females who were distracted, in particular as males might overestimate their smartphone, iPad use.

Additional comments

I still think that the study is interesting and authors could improve the paper by discussing the critical aspects I mentioned in the areas above.

Reviewer 2 ·

Basic reporting

The article is clear but there are some reporting issues: 1) the reference section does not follow the norms stated by this journal (e.g., not sorted by authors; the year should follow the authors’ names, etc.); 2) the references in the text also do not follow the norms (the name of the authors and the date should be stated). Also, sometimes the referred literature does not seem to be directly relevant for the article, especially in the Discussion (e.g., for me, the link between the distractions in classrooms and Khan Academy and MOOCs is not clear and I do not think it became clear after reading the discussion of this article, even though the discussion begins by mentioning Khan Academy).

Experimental design

The methods used in this article do not follow an experimental design. And this is actually my main concern about this paper. The authors simply passed a questionnaire after a lecture. This questionnaire had questions designed to measure learning of concepts from the lecture and also questions designed to measure the use of electronic devices during the lecture. Although it departs from an interesting question, the presented methods are not experimental, in the sense that use of electronic devices was not manipulated. And using this type of quasi-experimental design would be okay, especially in an exploratory study; but the measure of use of electronic devices rely only on self-report. I would like to see a better measure of use of electronic devices (e.g. time spent checking email; time on Facebook; etc.).

Validity of the findings

The validity of findings also raised some concerns. This could be treated as a correlational study but no correlational measures were presented. Also, only percentages are presented; I missed seeing some statistical analysis. Related to this, only 27 students were quizzed, so, even if the authors used an experimental design, this study would be underpowered. Therefore I do not think I can pronounce myself about the validity of these results.

Additional comments

I find the idea of testing the effects of distractions caused by electronic devices on student performance very interesting. Moreover, I think a survey to understand the percentages of students who use electronic devices during lectures is a good way to start. However, the findings reported here were obtained in only a lecture and with only 27 3rd year students from a School of Medicine. I think some statistical analysis and a second experiment (with variable manipulation or better measurement) would be a good way to continue this project. Also, extending this to other student populations could be interesting and lead to different results.

---

## Round 0.2 · Minor Revisions

Thank you for resubmitting and all the changes. Below are some final comments and once these are addressed can be accepted for publication.

Please include that this is a quasi-experimental pilot study with a small sample in the abstract and method and include all sample sizes in abstract.

All references to interactive vs traditional teaching should be removed or only discussed briefly in the future discussion section. The paper did not evaluate this topic. Saying "Educational institutions should perform thorough cost-benefit analysis, including evaluation of educational outcomes, before abandoning traditional lecture for modern educational strategies" is not relevant. A more suitable ending to the abstract is something like (dont have to use but):"However, results should be interpreted with caution as this this study was a small quasi-experimental design and further research should examine the influence of different types of distraction on different types of learning"

Similarly the paragraph below is not entirely relevant but okay if there is much more background on similar studies: (e.g. http://online.liebertpub.com/doi/abs/10.1089/cpb.2008.0107 ) which are attempting to understand a similar mechanism.

"Existing research has shown that by exhausting attention capacity through task-relevant processing, you can reduce the processing of task-irrelevant information (Forster & Lavie, 2009). ......"

"This paper reports outcomes of a small study that was designed to evaluate the learning outcomes of a traditional lecture among Gen Y students" NOTE:...This is not evaluating the learning outcomes of a traditional lecture. It is assessing the role of distraction vs not in learning outcomes of a single lecture.

Also - please note if the survey was anonymous or not.

Thank you.

---

## Round 0.3 · accepted · Accept

Thank you for making the changes. It is a nice pilot study and help us understand the mechanisms of distraction in future studies. Please make sure the manuscript is within PeerJ editorial guidelines. Good luck.